# Extracellular Vesicles of Patients on Peritoneal Dialysis Inhibit the TGF-β- and PDGF-B-Mediated Fibrotic Processes

**DOI:** 10.3390/cells13070605

**Published:** 2024-03-29

**Authors:** Beáta Szebeni, Apor Veres-Székely, Domonkos Pap, Péter Bokrossy, Zoltán Varga, Anikó Gaál, Judith Mihály, Éva Pállinger, István M. Takács, Csenge Pajtók, Mária Bernáth, György S. Reusz, Attila J. Szabó, Ádám Vannay

**Affiliations:** 1Pediatric Center, MTA Center of Excellence, Semmelweis University, 1083 Budapest, Hungary; 2HUN-REN–SU Pediatrics and Nephrology Research Group, 1052 Budapest, Hungary; 3TTK Biological Nanochemistry Research Group, Institute of Materials and Environmental Chemistry, Research Centre for Natural Sciences, 1117 Budapest, Hungary; 4Department of Genetics, Cell- and Immunobiology, Semmelweis University, 1089 Budapest, Hungary

**Keywords:** extracellular vesicles, peritoneal dialysis, fibrosis, mesenchymal transition, therapy

## Abstract

Among patients on peritoneal dialysis (PD), 50–80% will develop peritoneal fibrosis, and 0.5–4.4% will develop life-threatening encapsulating peritoneal sclerosis (EPS). Here, we investigated the role of extracellular vesicles (EVs) on the TGF-β- and PDGF-B-driven processes of peritoneal fibrosis. EVs were isolated from the peritoneal dialysis effluent (PDE) of children receiving continuous ambulatory PD. The impact of PDE-EVs on the epithelial–mesenchymal transition (EMT) and collagen production of the peritoneal mesothelial cells and fibroblasts were investigated in vitro and in vivo in the chlorhexidine digluconate (CG)-induced mice model of peritoneal fibrosis. PDE-EVs showed spherical morphology in the 100 nm size range, and their spectral features, CD63, and annexin positivity were characteristic of EVs. PDE-EVs penetrated into the peritoneal mesothelial cells and fibroblasts and reduced their PDE- or PDGF-B-induced proliferation. Furthermore, PDE-EVs inhibited the PDE- or TGF-β-induced EMT and collagen production of the investigated cell types. PDE-EVs contributed to the mesothelial layer integrity and decreased the submesothelial thickening of CG-treated mice. We demonstrated that PDE-EVs significantly inhibit the PDGF-B- or TGF-β-induced fibrotic processes in vitro and in vivo, suggesting that EVs may contribute to new therapeutic strategies to treat peritoneal fibrosis and other fibroproliferative diseases.

## 1. Introduction

The incidence of fibroproliferative diseases (FDs) is continuously increasing and has become a major health problem responsible for almost every second death in the developed world [1,2]. The common hallmark of FDs is the activation of fibroblasts and the excessive accumulation of the extracellular matrix (ECM) leading to the destruction of the healthy architecture of the affected organ and finally to the decline of its function [2].

Peritoneal dialysis (PD) using the peritoneum as a dialyzing membrane to exchange water and solutes is a widely used renal replacement therapy. However, signs of peritoneal fibrosis develop in 50–80% of patients on PD within two years [3]. The main characteristics of peritoneal fibrosis are the loss of peritoneal mesothelial cells (MCs), the increased proliferation of α-smooth muscle actin (α-SMA)-positive fibroblasts, and the consequent excessive accumulation of ECM [4,5]. All these processes finally lead to the progressive submesothelial thickening and impaired peritoneal filtration function, making it necessary to change the modality of the renal replacement therapy [4]. However, in severe cases, life-threatening encapsulating peritoneal sclerosis (EPS) may develop, characterized by peritoneal thickening, ultrafiltration failure (UFF), and intestinal obstruction [6], even after the patient is transferred for hemodialysis or after a renal transplant [7,8]. Currently, therapeutic possibilities for delaying and even reversing the peritoneal fibrosis, similarly to the treatment of other FDs, are fairly limited.

Extracellular vesicles (EVs) are lipid bilayer-coated biological nanoparticles released by different cell types [9,10]. The key function of EVs is related to the transfer of their molecular cargo into the neighboring cells, which has been suggested to modify the biological processes of the recipient cells [10,11,12,13]. Only a few previous studies have identified EVs in the peritoneal dialysate (PDE) to date. Indeed, the proteomic and bioinformatic studies of Carreras-Planella et al. and Pearson et al. suggested that the analysis of PDE-EVs could represent a potential non-invasive method for the early detection of peritoneal membrane damage in PD patients [14,15]. Other studies by Bruschi et al. and Corciulo et al. demonstrated that PDE-EVs have a mesothelial origin [16,17] and suggested that their water channel aquaporin 1 content is positively correlated with the remaining ultrafiltration capacity of the peritoneal membrane [17]. However, none of these studies investigated the potential role of PDE-EVs on the pathomechanism of peritoneal fibrosis.

It is known in the literature that there are EVs in PDE, but their biological role was previously unknown. Since the most important complication of PD is peritoneal scarring, we wanted to know whether this process is affected by EVs. Therefore, we investigated whether PDE-EVs could affect fibrosis-related processes, including the mesenchymal transition of mesothelial cells and/or the proliferation and ECM production of activated fibroblasts resulting in the reduced submesothelial thickening of the peritoneum in vivo.

## 2. Materials and Methods

### 2.1. Human Samples

The PDE of patients A–I and peritoneal biopsies of patients B, E, and J were collected from children on PD in the Pediatric Center, Semmelweis University (31224-5/2017/EKU). Clinical characteristics of patients enrolled in analyses are described below (Table 1). Eight male and two female patients were enrolled in the study. Their mean age was 13.8 ± 9.3 years, and the duration of their PD treatment was 36.8 ± 35.9 months. Five patients had kidney and urinary tract malformation, three patients suffered from cortical/tubular necrosis, one patient suffered from glomerulopathy, and one patient had cystinosis. The first daytime PDEs of patients, dialyzed with 1.5% glucose-containing PD solution (Fresenius Medical Care, London, UK), were collected for the experiments. PDEs were immediately centrifuged to separate the cells from the supernatants. Both the collected cells as well as the cell-free PDEs were further processed to isolate primary peritoneal fibroblasts (PDE-FBs) or EVs, respectively. A part of the cell-free PDE was also stored at −80 °C for further molecular biological measurements or in vitro experiments.

Parietal peritoneal biopsies were collected at the time of insertion (for primary peritoneal mesothelial cells, labeled P-MCs) or at the time of the removal of the Tenckhoff catheter (for primary peritoneal fibroblasts, labeled P-FB). Primary cells were isolated and cultured according to the protocols below.

During the in vitro experiments, P-MCs and PDE-FBs were treated with the PDE of patient G and with that of patient I, whilst P-FBs were treated with the PDE of patient G or with that of patient I, unless indicated otherwise. The detailed description of the experimental layouts is shown in Appendix A.

### 2.2. Lyophilization of PDE Samples

As described previously, the PDE samples were snap-frozen to the wall of a 50 mL Falcon centrifuge tube by spinning in liquid nitrogen to increase the drying surface spinning in liquid nitrogen to increase the drying surface [18]. Freeze-drying and the setup of lyophilization protocol were performed with a ScanVac CoolSafe Touch Superior device (LaboGene A/S, Allerod, Denmark).

### 2.3. Preparation of Human Peritoneal Mesothelial Cells and Fibroblasts

P-MCs were isolated from the peritoneal biopsy of patient J. Briefly, the biopsy was minced and digested in 0.25% trypsin-EDTA (Thermo Fisher Scientific, Waltham, MA, USA) at 37 °C for 30 min; then, after several washing steps with PBS, the sample was centrifuged at 160× *g* for 10 min, and the resulting pellet was then seeded into a 6-well cell culture plate (Sarstedt, Nünbrecht, Germany) and the cells were let to grow in M199 medium supplemented with 10% heat-inactivated fetal calf serum (FCS, Thermo Fisher Scientific, Waltham, MA, USA), 400 nM hydrocortisone (Thermo Fisher Scientific, USA), 870 nM insulin (Thermo Fisher Scientific, USA), 20 mM HEPES (Thermo Fisher Scientific, USA), 3.3 nM epithelial growth factor (EGF, R&D, Inc., Minneapolis, MN, USA), and 1% penicillin and streptomycin (Thermo Fisher Scientific, USA), at 37 °C in a humidified atmosphere containing 5% CO_2_ in collagen-coated flasks (Sarstedt, Germany). PDE-FBs were isolated from the PDE samples of patients A, F, G, and I or from the peritoneal biopsies of patient B or E (P-FBs) (Table 1).

Briefly, the PDE samples of patients A, F, G, and I were centrifuged at 160× *g* for 20 min at room temperature (Rotanta 460R, Hettice Zentrifugen, Tuttingen, Germany) immediately after sampling. The pelleted cells were grown in Dulbecco’s modified Eagle’s medium/Nutrient Mixture F-12 (DMEM-F12, Thermo Fisher Scientific, USA) medium supplemented with 10% FCS, and 1% penicillin and streptomycin at 37 °C in a humidified atmosphere containing 5% CO_2_ in collagen-coated flasks (Sarstedt, Germany). The experiments were performed on the PDE-FBs of patient G, unless indicated otherwise (Appendix A).

P-FBs were isolated from peritoneal biopsies of patient B or E. The biopsies were minced and digested in 1 mg/mL collagenase type II solution (Thermo Fisher Scientific, USA) at 37 °C for 30 min, and then, after several washing steps with PBS, the samples were centrifuged at 160× *g* for 10 min (Rotanta 460R, Hettice Zentrifugen, Germany). The resulting pellets were seeded into 6-well cell culture plates (Sarstedt, Germany) and the cells were let to grow in DMEM-F12, identically like PDE-FBs. The experiments were performed on the P-FBs of patient B, unless indicated otherwise (Appendix A). A detailed description of the experimental layouts is shown in Appendix A.

### 2.4. Isolation of PDE-Derived Extracellular Vesicle (PDE-EV)

Three hundred milliliters of PDE samples from patients A, F, G, and I were immediately centrifuged at 160× *g* for 20 min at RT (Rotanta 460R, Hettice Zentrifugen, Germany) and their cell-free supernatants were passed through a 0.2 µm syringe filter (Millipore Express^®^ PLUS, Merck, Darmstadt, Germany). The resulting solutions (250 mL) were first concentrated using a stirred-cell ultrafiltration device (Millipore, Germany) equipped with a 100 kDa MWCO polyethersulfone ultrafiltration disc (Millipore, Germany), and then using 100 kDa MWCO centrifugation filters (Amicon^®^ Ultra-15 Centrifugal Filter Unit, Merck, Germany) at 18,000× *g* for 5 min (Rotanta 460R, Hettice Zentrifugen, Germany). Finally, the concentrated samples were loaded onto a size-exclusion chromatography column (Sepharose CL-2B, Merck, Germany) to purify EVs. The PBS soluble fractions, namely PDE-EVs, were immediately further analyzed, avoiding the unnecessary freeze–thaw cycles for validating their EV content. As a first step, the particle numbers and size distributions of the PDE-EVs were determined by microfluidic resistive pulse sensing (MPRS) and dynamic light scattering (DLS). Subsequently, freeze-fracture combined transmission electron microscopy was carried out to visualize their spherical morphology, and Fourier transform infrared spectroscopy was used to determine their protein lipid contents and ratios. The presence of surface markers from the tetraspanin family (CD9, CD63, and CD81) as well as the annexin positivity and calnexin (endoplasmic reticulum marker) negativity of these EVs was also investigated by flow cytometric analysis. The bicinchoninic acid (BCA) protein assay was used to determine their total protein concentrations. Following their characterization, EVs were immediately (freshly) used in the in vitro or in vivo experiments.

During the in vitro experiments, the P-MCs and PDE-FBs were treated with the PDE-EVs of patient G and P-FBs were treated with the PDE-EVs of patient G or I, unless indicated otherwise. For an easier understanding, the exact experimental layouts are outlined in Appendix A.

### 2.5. Microfluidic Resistive Pulse Sensing

The particle numbers and size distributions of PDE-EVs were determined by MPRS as a first step. Briefly, PDE-EVs were diluted 10-fold with 1 mg/mL bovine serum albumin (BSA, Sigma-Aldrich, Hamburg, Germany) dissolved in PBS buffer (Sigma-Aldrich, Germany) and filtered through a VivaSpin 500–100 kDa MWCO membrane filter (Sartorius, Ulm, Germany) according to the manufacturer’s instructions. Measurements were performed using factory-calibrated TS-400, TS-900, and TS-2000 cartridges (Spectradyne LLC, Signal Hill CA, USA), which cover a measurement range of particles from 65 nm to 2000 nm.

### 2.6. Dynamic Light Scattering

Besides MRPS, size distribution was also determined by DLS. Briefly, 100 μL of 1.5 × 10^11^ particles/mL PDE-EVs were used in low-volume disposable plastic cuvettes (UVette; Eppendorf Austria,). Hydrodynamic diameter of PDE-EVs was measured by W130i DLS instrument (AvidNano, High Wycombe, UK), and data evaluation was performed with iSize 3.0 software (AvidNano, UK).

### 2.7. Freeze-Fracture Combined Transmission Electron Microscopy

PDE-EV samples were mixed with glycerol (Sigma-Aldrich, Germany) in 3:1 sample-to-glycerol volume ratio as cryoprotectant. A 2 μl sample of 1.5 × 10^11^ particles/mL PDE-EVs was pipetted onto a gold sample holder and frozen by immediately placing them into partially solidified Freon for 20 s. Fracturing was performed at −100 °C in a Balzers freeze-fracture device (Balzers BAF 400D, Balzers AG, Liechtenstein). The replicas of the fractured surfaces were made by platinum–carbon evaporation and then cleaned with a water solution of surfactant and washed with distilled water. The platinum–carbon replicas were placed on 200-mesh copper grids and examined with a MORGAGNI 268D (FEI, Eindhoven, The Netherlands) transmission electron microscope.

### 2.8. Fourier Transform Infrared Spectroscopy

FTIR measurements were carried out using a Varian 2000 spectrometer (Scimitar Series, USA) fitted with a diamond-attenuated total reflection cell (‘Golden Gate’ single-reflection attenuated total reflectance (ATR) unit, Specac, United Kingdom). Five microliters of 1.5 × 10^11^ particles/mL PDE-EVs was pipetted onto the diamond ATR surface and a thin dry film was obtained by slowly evaporating the solvent under ambient conditions (approx. 10 min). Typically, 64 scans were collected at a nominal resolution of 2 cm^−1^. ATR correction, buffer background spectral subtraction, and other spectral evaluations were performed with the Grams/32 software package (Galactic Inc., USA).

### 2.9. Flow Cytometry

The surface markers of the PDE-EVs (tetraspanins, CD9, CD63, and CD81) as well as annexin positivity and calnexin negativity were determined by flow cytometry using FACS Calibur (BD Biosciences, San Jose, CA, USA). Thirty microliters of 1.5 × 10^11^ particles/mL PDE-EVs was first adsorbed onto the surface of formaldehyde/sulfate Latex-Beads (Molecular Probes, Eugene, OR, USA). In brief, EVs were loaded onto 3.8 μm diameter 4.0 *v*/*v*% beads dissolved in 100 μL phosphate-buffered saline (PBS) and incubated overnight at 4 °C under gentle agitation. PBS containing 2% BSA was used as a negative control. Samples were incubated for 1 h in PBS containing 2% BSA; then, 300 μL 0.9% NaCl solution was added, and finally, the samples were centrifuged for 10 min at 400× *g*. The beads were resuspended in 200 μL 100 mM glycerol solution and incubated for 30 min at RT. Samples were then labeled with specific antibodies against CD9 ((fluorescein (FITC), Sigma;#:SAB4700095)), CD63 ((phycoerythrin (PE), Sigma;#:SAB4700218)), CD81 (FITC, Molecular Probes;#:A15753), annexin V (FITC, SONY; #:3804530), and calnexin (E-10, Santa Cruz Biotechnology, Dallas, USA) for 24 h at 4 °C. For data analysis, FlowJo Software v9(Tri Star Inc., Ashland, OR, USA) was used.

### 2.10. Protein Content

To measure the total protein content of the PDE-EV samples, they were lysed in lysis buffer (1% Triton X-100 + 0.1% SDS in H_2_O) for 30 min in ice. Protein quantification was performed using the Pierce^TM^ BCA Protein Assay kit (Thermo Fisher Scientific, USA) according to the manufacturer’s instructions. Samples were measured in triplicate and values were averaged for total protein estimation.

### 2.11. Labeling of PDE-EV

To test the in vitro and in vivo internalization of EVs, they were labeled by DiI (DiIC18(3) (1,1′-dioctadecyl-3,3,3′,3′-tetramethylindocarbocyanine perchlorate, Molecular Probes, Thermo Fisher Scientific, USA). Briefly, 100 µL of 1.5 × 10^11^ particles/mL PDE-EVs were incubated with 2 µL of 1 mg/mL DiI dye for 30 min, 37 °C. The free dye was separated from the DiI-labeled EVs by washing with 400 µL PBS on size exclusion chromatography columns (Sepharose CL-2B, Merck, Germany). PBS containing 1 mg/mL DiI in the absence of EVs was also loaded onto Sepharose columns and washed with PBS, and these latter samples were used as negative controls.

### 2.12. PDE-EV Internalization Experiment

We seeded 6 × 10^4^ P-MCs, PDE-FBs or P-FBs in 4-well cell culture chambers (Sarstedt, Germany) to reach approximately 80% confluence. Cells were then incubated with cell culture medium containing DiI-labeled PDE-EVs (4.5 × 10^10^ particles/mL) or PBS (see above) for 24 h at 37 °C in a humidified atmosphere of 5% CO_2_. Subsequently, after several washes with PBS, cell nuclei were stained with Hoechst 33342 (1:2000, Merck, Germany) and then the slides were cover-slipped with ProLong^TM^ Gold Antifade reagent (Thermo Fisher Scientific, USA). To visualize the internalization of the EVs, an Olympus IX81 microscope was used (Olympus Corporation, Tokyo, Japan).

### 2.13. Western Blot

The lyophilized PDE samples from patients A–I or the PDE-EVs of patient G were homogenized in lysis buffer containing 50 mM 4-(2-hydroxyethyl)-1-piperazineethanesulfonic acid (HEPES), 150 mM sodium chloride (NaCl), 1% Triton X-100, 5 mM edetic acid (EDTA), 5 mM egtazic acid (EGTA), 20 mM sodium pyrophosphate (Na_4_P_2_O_7_), 20 mM sodium fluoride (NaF), 0.2 mg/mL phenylmethylsulfonyl fluoride (PMSF), 0.01 mg/mL leupeptin, and 0.01 mg/mL aprotinin (each substance was obtained from Merck, Germany). The homogenized samples were denatured in 4× Laemli buffer at 95 °C for 5 min. Denatured samples (10 μg protein/lane) were separated on 4–20% gradient SDS polyacrylamide gel and then transferred into nitrocellulose membranes (Bio-Rad, Hercules, CA USA). The nitrocellulose membranes were blocked with 5% non-fat milk in TRIS-buffered saline (TBS) for 1 h at room temperature (RT). Thereafter, they were incubated overnight at 4 °C with antibodies specific for platelet-derived growth factor B (PDGF-B) (#C5, Santa Cruz Biotechnology, USA), transforming growth factor-beta (TGF-β)1/2/3 (H-112; Santa Cruz Biotechnology, USA), glyceraldehyde-3-phosphate dehydrogenase (GAPDH) (#sc-47724; 1:2000, Santa Cruz Biotechnology, USA), heat shock protein 70 (HSP70) (#4872, Cell Signaling Technology, UK), E-cadherin (#ab76055, Abcam, UK), or cytokeratin-18 (CK-18) (sc-31700, Santa Cruz Biotechnology, USA). After repeatedly washing with TBS containing 0.05% Tween-20 and 1% non-fat milk, membranes were incubated with the corresponding horseradish peroxidase-conjugated rabbit (#A0545, Sigma-Aldrich), mouse, or goat (#G-21040, #A11055, Thermo Fisher Scientific, USA) secondary antibodies for 1 h at RT. Bands of interest were detected using an enhanced chemiluminescence detection reagent (Western Blotting Luminol Reagent, GE Healthcare, Waukesha, WI) and quantified by densitometry (VersaDoc, Quantity One Analysis software; Bio-Rad, USA) as integrated optical density after background subtraction. Relative protein levels were determined by a comparison with GAPDH as an internal control. Data were normalized and presented as the ratio of their control group value.

### 2.14. MTT (Cell Proliferation) Assay

P-MCs, PDE-FBs, and P-FBs were seeded into 96-well plates at a density of 4 × 10^3^ cells/well. MTT assays were performed on 30 *v*/*v*% PDE, 10 ng/mL PDGF-B (R&D Systems, USA), or 10 ng/mL TGF-β (R&D Systems, USA)-treated P-MCs, PDE-FBs, or P-FBs (n = 5 well/treatment group) in the absence or presence of 4.5 × 10^9^–10^10^ particles/mL PDE-EVs according to the instructions of the manufacturer (MTT Cell Proliferation/Viability Assay kit, R&D Systems, USA). Absorbance was recorded at 570 nm and at 690 nm as background in a SPECTROstar Nano microplate reader using SPECTROstar Nano MARS v3.32 software (BMG Labtech, Ortenberg, Germany). Vehicle-treated cells (4 mM hydrogen chloride (HCl) in case of PDGF, TGF-β, 30 *v*/*v*% 1.5% glucose-containing peritoneal dialysis fluid (Fresenius Medical Care, UK) in the case of PDE and PBS in the case of EV) served as controls. Results were normalized and determined as the percentage ratio of control group values.

### 2.15. Sirius Red Assay (Total Collagen Production)

To investigate the amount of fibrillar collagens, Sirius Red assay was performed (n = 5-well/treatment group). After 48 h of 30 *v*/*v*% PDE, 10 ng/mL TGF-β or 10 ng/mL PDGF-B treatment in the presence or absence of 4.5 × 10^9^–10^10^ particles/mL EVs P-MCs, PDE-FBs, or P-FBs were incubated in a fixative solution containing 26% ethanol, 3.7% formaldehyde, 2% glacial acetic acid for 15 min at room temperature. Samples were stained for 1 h at room temperature with a 0.1% solution of Sirius Red (DirectRed80) dissolved in 1% acetic acid, and then washed three times with 200 μL of 0.1 M HCl, and finally the bounded dye was dissolved by adding 100 μL of 0.1 M NaOH (all reagents were purchased from Merck, Germany). Absorbance was recorded at 544 nm and at 690 nm as background in a SPECTROstar Nano microplate reader using SPECTROstar Nano MARS v3.32 software (BMG Labtech, Germany). Vehicle-treated (4 mM HCl in the case of TGF-β, 30% PDF the in case of PDE, and PBS in case of EV) cells served as controls. Results were normalized and determined as a percentage ratio of the control group values.

### 2.16. Mouse Model of Peritoneal Fibrosis

All experiments were approved by the institutional committee on animal welfare (PE/EA/361-2/2019). Experiments were performed on 7–8-week-old male C57Bl/6J mice (Charles River Laboratories, Germany). Animals were housed in a temperature-controlled (22 ± 1 °C) room with alternating light and dark cycles and had free access to standard chow and water. Peritoneal fibrosis was induced by the intraperitoneal (i.p.) injection of 0.3 mL 0.1% chlorhexidine digluconate (CG; Merck, Germany) diluted in 15% ethanol and 85% phosphate-buffered saline (PBS) every day for one week (CG treated group, n = 8) [19]. The control group of mice received daily 0.3 mL PBS i.p. (PBS group, n = 8). A group of CG and PBS-treated mice were also treated with the 50 µg PDE-EV of patient F dissolved in 0.3 mL PBS on the 1st and 4th day of the experiment.

To determine the ultrafiltration capacity of the peritoneum, mice were given 0.2 mL 10 mg/mL.

On the 3rd and 7th day of the experiment, 70 kDa tetramethyl–rhodamine isothiocyanate (TRITC) and 0.2 mL 10 mg/mL 4 KDa FITC-labeled dextran were administered i.p. (Merck, Germany). Thirty minutes later, blood samples were taken from their tail vein using capillaries and the fluorescent signal of dextran was investigated using a SPECTROstar Nano microplate reader using SPECTROstar Nano MARS v3.32 software (BMG Labtech, Germany). The mice were sacrificed on the 7th day, and the parietal peritoneal samples were removed and fixed in 4% buffered formaldehyde or immediately snap-frozen and stored at −80 °C until further molecular biological measurements.

### 2.17. Real-Time Reverse Transcriptase-Polymerase Chain Reaction (qRT-PCR)

Total RNA was isolated by Geneaid Total RNA Mini Kit (Geneaid Biotech Ltd., New Taipei City, Taiwan). The RNA (500 ng/sample) was reverse-transcribed using SuperScript III reverse transcriptase (Thermo Fisher Scientific, USA) to generate first-stranded cDNA. The mRNA expression of the target molecules was determined by real-time PCR using SYBR Green PCR master mix on a Light Cycler 480 system (Roche Diagnostics, Mannheim, Germany). The nucleotide sequences and species specificity of the applied primer pairs and the lengths of the resulted PCR products are shown in Table 2. The relative mRNA expressions were analyzed by Light-Cycler 480 software version 1.5.0.39 and determined by comparison with *GAPDH* as an internal control. Data were normalized and presented as the ratio of their control values.

### 2.18. Masson’s Trichrome Staining

The 4 μm thick paraformaldehyde-fixed (4%, pH 7.4) and paraffin-embedded sections were stained with Masson’s trichrome stain. For the evaluation of peritoneal fibrosis, a non-overlapping image of Masson’s trichrome section was taken by an Olympus IX81 microscope system (Olympus Corporation, Tokyo, Japan) using the ×20 objective. The thickness of the submesothelial region was quantitatively measured using Image J software (National Institute of Health, USA) by drawing a line around the perimeter of the thickened area.

### 2.19. Immunofluorescent Staining

Primary cells were identified as P-MCs, PDE-FBs, and P-FBs based on their morphology, α-SMA (#A2547, Merck, Germany), fibronectin (#ab2413, Abcam, UK), vimentin (#sc-7557, Santa Cruz Biotechnology, USA), and CK-18 (sc-31700, Santa Cruz Biotechnology, USA) positivity. After fixation by methanol for ten minutes at RT, chambers were first incubated with the appropriate primary antibody for 60 min at RT; then, with the appropriate Alexa Fluor^®^ conjugated secondary antibodies (#A10680, #A21206, #A11057, Thermo Fisher Scientific, USA) for 60 min at RT. Nuclei were stained with Hoechst 33342 (Merck, Germany). Finally, the slides were cover-slipped with ProLongTM Gold Antifade reagent (Thermo Fisher Scientific, USA). The appropriate controls were performed by omitting the primary antibodies.

The fresh frozen OCT (Tissue-Tek O.C.T, Quiagen, Venlo, The Netherlands) embedded peritoneal samples cut into 4 µm sections. Sections were incubated with specific primary antibodies specific for CK-18, vimentin, α-SMA, or COL1A1 (#sc-293182, Santa Cruz Biotechnology, USA), then washed and incubated with the appropriate Alexa Fluor^®^ conjugated secondary antibodies (same as we described above, Thermo Fisher Scientific, USA). Nuclei were stained with Hoechst 33342 and slides were cover-slipped with ProLongTM Gold Antifade reagent (Thermo Fisher Scientific, USA). To visualize the stained sections, an Olympus IX81 microscope system was used (Olympus Corporation, Tokyo, Japan).

### 2.20. Gene Ontology (GO) Term Analysis

GO Enrichment Analysis (powered by PANTHER) was carried out to find the overrepresented GO terms using annotations of PDE-EV proteins based on the previous proteomic data presented by Pearson and coworkers [14]. The GO terms for Homo sapiens were included in the analysis.

### 2.21. Statistical Analysis

Data were analyzed using GraphPad Prism 8.0. software (GraphPad Software Inc., La Jolla, CA, USA). After testing the normality with Kolmogorov–Smirnov test, one-way or two-way ANOVA with Bonferroni test was used to compare the difference between the respective groups. Values were expressed as a mean ± SD. Correlation was determined by Pearson’s coefficients (r). The applied tests, significances, and number of elements (n) are indicated in each Figure legend.

## 3. Results

### 3.1. PDGF-B and TGF-β Content of PDE

The PDGF-B and TGF-β content of PDEs were determined by Western blot (Figure 1a). Both factors were present in the PDEs of children with PD, and there was a strong positive correlation between the protein level of PDGF-B and TGF-β in the investigated samples (r = 0.870, *p* = 0.0045) (Figure 1b). In addition, a strong positive correlation was found between the duration of PD and the amount of PDGF-B (r = 0.762, *p* = 0.002) (Figure 1c); however, TGF-β protein level showed only a tendency to relate with a duration of PD treatment (r = 0.670, *p* = 0.055) (Figure 1d).

### 3.2. Characterization of PDE-EVs

Morphological characterization of PDE-EVs was performed by FF-TEM, which enabled the observation of the native structure of EVs. FF-TEM images demonstrated a spherical EV morphology (Figure 2a). According to the DLS measurements, the mean hydrodynamic diameter of the PDE-EVs was 186 nm with a standard deviation of 61 nm (Figure 2b). Similarly, according to the power law distribution, the MPRS measurements revealed that the size of the majority of EVs was in the sub-300 nm range with a concentration of 1.5 × 10^11^ particles/mL, but EVs smaller than 100 nm were also present in the samples (Figure 2c). The total protein contents of PDE-EVs (from patients A, F, G, I), was 547 ± 60.4 µg/mL as determined by BCA assay. After buffer subtraction, the infrared (IR) spectrum of PDE-EVs showed typical spectral features characteristic for EVs (Figure 2d). Bands at 3290, 1657, and 1545 cm^−1^ belongs to Amide A, Amide I, and Amide II vibrations of the peptide backbone. The acyl chains of the lipid bilayer can be featured by the asymmetric and symmetric stretching (ν_as_ CH_2_ at 2918 cm^−1^ and ν_s_ CH_2_ at 2850 cm^−1^). As we established in a previous paper [20], the ratio of the area of amide I band and that of the C–H stretching regio (from 3020 to 2800 cm^−1^) provides a ‘spectroscopic protein-to-lipid’ ratio and serves as a quality control of EV samples. For PDE-EV samples, this value is 0.755 and fits well to common P/Lspectr value of EVs (e.g., human blood serum-derived EVs usually show P/Lspectr = 0.65 ± 0.1) [20]. Flow cytometric analysis demonstrated annexin-V (phospholipid layer marker), CD9, and CD63 (EV surface marker) positivity of the PDE-EVs (Figure 2e). PDE-EVs were negative for CD81 and calnexin, which are markers of endoplasmic reticulum (Figure 2e). The cargo of the PDE-EVs contained a heat shock protein (HSP) 70, cytokeratin (CK) 18, and E-cadherin supporting their mesothelial origin (Figure 2f).

### 3.3. Characterization of the Different Primary Cells and Their PDE-EV Uptakes

The P-MCs had a typical ‘cobblestone’-like monolayer appearance and were positive for CK-18, and negative for alpha-smooth muscle actin (α-SMA) (Figure 3). The PDE-FBs were identified as fibroblasts based on their spindle-like morphology, α-SMA, fibronectin, and vimentin positivity. These cells also expressed CK-18, suggesting their mesothelial origin (Figure 3). P-FBs showed similar morphology to PDE-FBs; however, these cells were negative for CK-18 (Figure 3). The DiI-labeled PDE-EVs penetrated into each type of primary cell (Figure 3).

### 3.4. PDE-EVs Inhibited the PDE- or PDGF-B-Induced Proliferation of P-MCs PDE-FBs and P-FBs

PDE-EV treatment decreased both the endogenous as well as the PDE- or PDGF-B-induced proliferation of P-MCs (Figure 4a,c), PDE-FBs (Figure 4e,g), and P-FBs (Figure 4k). The PDE-EV treatment also decreased the MKI67 expression of PDE- or PDGF-B-treated cells (Figure 4b,d,f,h,j,l). To further evaluate whether the effects of PDE-EVs depend on the PD patients from whom the PDEs, primary cells or EVs were isolated, primary P-MCs, PDE-FBs, and P-FBs were treated with the PDE and with the PDE-EVs of different patients (Appendix A). However, the origin of the cells or EVs did not influence the results, and EVs from any patient inhibited the proliferation of primary cells of any patient (Appendix A). TGF-β1 treatment had no effect on the proliferation of either the P-MCs (patient J), PDE-FBs (patient G), or P-FBs (patient B) (Appendix A).

### 3.5. PDE-EVs Inhibited the PDE or TGF-β-Induced Mesenchymal Transition of P-MCs and the Collagen Production of P-MCs

PDE-treated P-MCs had a cobblestone-like morphology, strong CK-18 positivity compared to the control cells, and only a weak α-SMA expression (Figure 5a). The PDE treatment decreased the mRNA expression of tight junction gene claudin (CLDN) 1, and increased that of vimentin (VIM), fibronectin (FN), collagen-type I alpha 1 (COL1A1), and keratin (KRT) 18 of the P-MCs compared to the control cells (Figure 5b). After the PDE treatment, the total collagen production of the P-MCs increased, as shown by the SiriusRed assay (Figure 5c). PDE-EVs inhibited the PDE-induced mesothelial mesenchymal transition (MMT) of P-MCs confirmed by the increased mRNA expression of tight junction genes including tight junction protein (TJP) 1 and CLDN1, and decreased the mRNA expression of VIM, FN, and COL1A1 compared to the PDE-treated cells (Figure 5b). Moreover, the PDE-EV treatment decreased the total collagen production of the PDE-treated P-MCs compared to the PDE-treated cells (Figure 5c). The TGF-β treatment induced a strong α-SMA positivity of P-MCs (Figure 5a). Moreover, the TGF-β induced the mRNA expression of VIM, FN, COL1A1, collagen type III alpha 1 (COL3A1) (Figure 5d), and the total collagen production (Figure 5e), but decreased the mRNA expression of KRT18 of the P-MCs compared to control cells (Figure 5d). Following the PDE-EV treatment, the α-SMA positivity of the TGF-β-treated P-MCs decreased (Figure 5a). Furthermore, PDE-EVs inhibited the TGF-β-induced expression of VIM, COL1A1, and COL3A1 of P-MCs (Figure 5d). The PDGF-B treatment increased the mRNA expression of KRT18 of P-MCs (Appendix A); however, it had no effect on α-SMA positivity (Appendix A), VIM, and FN mRNA expression (Appendix A), or the total collagen accumulation of the cells (Appendix A). Since PDGF-B treatment did not induce MMT or collagen production of P-MCs, the effect of PDE-EVs was not further investigated (Appendix A).

### 3.6. PDE-EVs Inhibit the PDE- or TGF-β-Induced Collagen Production of PDE-FBs and P-FBs

PDE treatment increased the mRNA expression of COL1A1 and COL3A1 as well as the total collagen production of PDE-FBs (Figure 6a). Similarly, PDE treatment induced the mRNA expression of COL1A1 and COL3A1, as well as the total collagen production of P-FBs (Figure 6c). PDE-EVs decreased the mRNA expression of COL1A1 and COL3A1 as well as the total collagen production of the PDE-treated PDE-FBs (Figure 6a). Similarly, the PDE-EVs decreased the mRNA expression of COL1A1 and the total collagen production of the PDE-treated P-FBs (Figure 6c). The TGF-β treatment increased the mRNA expression of COL1A1 and COL3A1 as well as the total collagen production of PDE-FBs (Figure 6b). In the case of P-FBs, the TGF-β treatment increased the total collagen production (Figure 6d). PDE-derived EVs inhibited the TGF-β treatment-induced expression of COL1A1, COL3A1, and the total collagen production of PDE-FBs and P-FBs (Figure 6b,d).

Since the PDGF-B treatment did not induce the collagen production of PDE-FBs and P-FBs, the effect of PDE-EVs was not further examined (Appendix A).

### 3.7. Effect of PDE-EVs on Peritoneal Scar Tissue Formation In Vivo

To investigate the in vivo effect of PDE-EVs, the most commonly used CG-induced mouse model of peritoneal fibrosis was adopted [21]. The intraperitoneally administrated DiI-labeled PDE-EVs resulted in a strong fluorescent signal in the parietal and visceral peritoneum of CG-treated mice, suggesting the significant peritoneal uptake of PDE-EVs (Figure 7b). The peritoneum of the control mice was CK-18 immunopositive, but negative for mesenchymal or fibrosis markers, including α-SMA, vimentin, fibronectin, and Col 1-α (Figure 8a). CG treatment increased the peritoneal mRNA expression of Mki67, Pdgfb, Acta2, and Vim, but decreased that of the Krt18 expression compared to the peritoneum of control mice (Figure 8b). In accordance, the peritoneum of CG-treated mice was positive for the mesenchymal and fibrosis markers, including the α-SMA, vimentin, fibronectin, and COL1-α, and it was negative for the CK-18 epithelial marker (Figure 8a). These changes suggest that the peritoneal mesothelial layer of the CG-treated mice that underwent MMT PDE-EV treatment increased the mRNA expression of Krt18 and decreased that of MKi67, Pdgfb, Acta2, and Vim (Figure 8b). Accordingly, the PDE-EV treatment of the mice almost completely inhibited the CG-induced immunohistological changes of the mesothelial layer, including its α-SMA, vimentin, and COL1-α positivity (Figure 8a). These molecular biological changes were in accordance with the histological changes of the peritoneal submesothelial zone (Figure 8c). Indeed, while the mean thickness of the submesothelial zone increased from 22.7 to 74.3 µm in response to CG treatment, it was almost completely normalized in response to the PDE-EV treatment (Figure 8d). Finally, we monitored the ultrafiltration capacity of the peritoneal membrane using tetramethylrhodamine isothiocyanate (TRITC)-labeled 70 kDa dextran and fluorescein-isothiocyanate-labeled 4 kDa dextrane. We found that, while CG treatment decreased the peritoneal ultrafiltration due to the submesothelial thickening [22] (Figure 7a and Figure 8e), the PDE-EV treatment slightly improved the peritoneal transport function on the third day of CG treatment (Figure 8e).

### 3.8. Role of PDE-EV Proteins on the Fibrotic Processes—Results of Gene Ontology Term Analysis

Gene Ontology (GO) Enrichment Analysis (powered by PANTHER) showed that seven of the PDE-EV proteins—based on the previous proteomic data of Pearson and coworkers [14]—were enriched in the GO term of the “negative regulation of fibroblast proliferation” (GO:0048147—consisting of 29 proteins) and 14 in the “ECM disassembly” (GO:0022617—consisting of 34 proteins) (Figure 9).

## 4. Discussion

In the present study, we examined the role of EVs on the mechanism of PD-related peritoneal fibrosis. Peritoneal fibrosis is characterized by the excessive accumulation of ECM components leading to the progressive thickening of the submesothelial compact zone of the peritoneal membrane [23,24]. In severe cases, life-threatening EPS, characterized by peritoneal sclerosis, and reduced ultrafiltration capacity, together with intestinal obstruction, may develop [6]. The reported average incidence of EPS is 0.5–4.4% in the USA [25]; however, it may reach 14–15% in patients who spent 5 or more years on PD [25,26]. The EPS represents a severe complication due to the high mortality rate, reaching 50% within 12 months from the diagnosis [8,27,28].

Our present study demonstrates that PDE-EVs, isolated from the PD effluent (PDE) of different children with PD, significantly inhibits the PDGF-B- or TGF-β-induced proliferation and/or the ECM production of P-MCs and fibroblasts of different origins. The PDE-EVs also penetrated into the peritoneum of mice and significantly reduced the chlorhexidine digluconate (CG)-induced peritoneal thickening in vivo.

First, in line with the others, we revealed the presence of the most relevant profibrotic factors, including PDGF-B and TGF-β [29] in the PDEs of children with PD (Figure 1). Despite the relatively low number of samples, the amount of PDGF-B strongly correlated with the duration of PD treatment. Moreover, the amount of TGF-β correlated with the amount of PDGF-B in the PDE, suggesting the involvement of both profibrotic factors in the pathomechanism of peritoneal fibrosis (Figure 1). These growth factors are known to play a central role in the scarring of almost all organs. Indeed, while PDGF-B is primarily responsible for proliferation, TGF-β has a strong impact on the ECM production of fibroblasts [30,31,32].

Upon the further examination the composition of PDE, we found that it also contains EVs in addition to profibrotic factors. According to our extensive analysis, the morphology, size distribution, as well as the protein-to-lipid ratio of the isolated PDE-EVs corresponded to typical EVs (Figure 2). Moreover, PDE-EVs were positive for the known EV-specific proteins, including annexin, CD9, CD63, HSP70, and for E-cadherin and CK-18, which is consistent with the previous work by Bruschi et al. [16], Corciulo et al. [17], and Huang et al. [33], suggesting that the isolated PDE-EVs are of the mesothelial origin (Figure 2).

Only a few previous studies [14,15,16,17] investigated some of the aspects of PDE-EVs, mainly focusing on their physical and molecular characteristics, but their biological role in the development of peritoneal fibrosis remained poorly understood. Indeed, Corciulo et al. provided evidence that the water channel aquaporin 1, playing a fundamental role in water ultrafiltration during peritoneal dialysis, is released into the PDE along with EVs of mesothelial origin [17]. Moreover, Huang et al. demonstrated that P-MC-derived EVs promote peritoneal fibrosis in mice receiving 4.25% glucose-containing PD fluid [33].

During PD, the released profibrotic factors [29,34] or EVs directly interact with the MCs of the peritoneum. Therefore, we first investigated the effect of PDE on the behavior of P-MCs isolated from the peritoneal biopsies of children with PD (Figure 3). We found that the PDE treatment increased the proliferation (Figure 4, Appendix A), the expression of mesenchymal markers, including vimentin and fibronectin (Figure 5), and decreased the expression of the TJ components of the P-MCs (Figure 5). However, the PDE-treated P-MCs remained positive for epithelial marker CK-18 and negative for the myofibroblast marker α-SMA (Figure 5). These data suggest that P-MCs undergo a partial epithelial–mesenchymal transition (EMT) after PDE treatment, when the cells enter into a hybrid state and simultaneously express both epithelial and mesenchymal markers. Previous studies have shown that epithelial cells of different origins can undergo a partial EMT when they acquire mesenchymal markers and express profibrotic factors, but may remain attached to their basement membrane due to their epithelial marker positivity [35,36]. Regarding the peritoneum, Patel et al. demonstrated that the adenoviral overexpression of PDGF-B led to the mesenchymal dedifferentiation of the MCs, which, however, remained in the mesothelial layer of the peritoneum [37]. They concluded that this “non-invasive” or partial EMT may have implications concerning peritoneal fibrosis.

To further clarify the molecular background of the partial EMT, the P-MCs were also treated with the most accepted profibrotic factors, including TGF-β and PDGF-B, present in the PDE of the children on PD.

We found that the TGF-β treatment increased the expression of mesenchymal markers and decreased the expression of the epithelial markers of P-MCs. Furthermore, these TGF-β-treated P-MCs increasingly produced collagens, consistent with their full EMT (Figure 5). In contrast, PDGF-B treatment induced the proliferation (Figure 4, Appendix A) and CK-18 expression of P-MCs without any effect on the expression of the mesenchymal markers (Appendix A). Taken together, these data suggest that the presence of PDGF-B in the PDE can alter the action of TGF-β, ultimately leading to the partial EMT of MCs.

It is believed that the primary function of the EVs is to participate in cell-to-cell communication [38,39,40]. The internalization of the EVs into the cells is thought to be essential in this process [11,12,13]. Thus, we investigated whether PDE-EVs can penetrate into the P-MCs (Figure 3). Thereafter, to further examine their effect, we demonstrated that the PDE-EV treatment inhibits the PDE- or TGF-β-induced mesenchymal differentiation and the collagen production of the P-MCs (Figure 5). Moreover, PDE-EVs also inhibited the PDE- or PDGF-B-induced proliferation of P-MCs (Figure 4). Taken together, these data suggest that PDE-EVs may inhibit TGF-β-induced dedifferentiation, collagen production, and the PDGF-B-induced proliferation of P-MCs, thus contributing to the maintenance of the healthy epithelial phenotype of MCs.

In the next series of in vitro experiments, we investigated whether there are mesothelial or other cells in the PDE of the patients with PD. Indeed, the patients’ PDEs contained spindle-like cells, which were immunopositive for mesenchymal as well as for epithelial markers, suggesting that these cells are of mesothelial origin (Figure 3). This observation, in accordance with our above experiment revealing the mesenchymal dedifferentiation of PDE-treated mesothelial cells (Figure 5), suggest that these cells are most likely mesothelial-derived fibroblasts (PDE-FBs). Some previous studies also identified the viable mesothelial or mesenchymal cells in the PDE of patients with PD. Indeed, Betjes et al. suggested that the increased presence of healthy viable mesothelial cells in the PDE may represent the increased turnover of the mesothelium, which could play a role in preventing the peritonitis of PD patients [41]. In accordance with our findings, others also revealed that these cells are rather dedifferentiated mesothelial cells, which are associated with the regeneration of the mesothelium and submesothelial region of the peritoneum [42,43]. However, there are only a few previous data about the possible role of PDE-FBs of mesothelial origin in the peritoneal regeneration or fibrosis.

Further investigating the behavior of these cells, we observed that, in response to PDE treatment, the PDE-FBs increasingly proliferate and produce ECM components (Figure 4). The similar behavior of PDE-FBs was observed in response to their PDGF-B or TGF-β treatment (Figure 4, Figure 5 and Figure 6), further supporting the fibroblast nature of these cells.

Thereafter, we studied the effect of PDE-EV on the behavior of PDE-FBs. In these experiments, we examined whether PDE-EVs—similarly to our previous experiment on P-MCs—are capable of penetrating into the cells (Figure 3), affecting the main functions of PDE-FBs.

We found that PDE-EVs entered into the cells and decreased their PDE- or TGF-β-induced ECM production (Figure 5), as well as their endogenous and PDE- or PDGF-B-induced proliferation (Figure 4). These experiments suggest that the PDE-EVs are capable of inhibiting the activation of fibroblasts and that, in accordance with our experiments on P-MCs, this effect of PDE-EVs is manifested at least in part through the inhibition of TGF-β- and/or PDGF-B-regulated processes.

In the last series of in vitro experiments, the effect of PDE-EV was also studied on the behavior of another type of fibroblasts on primary peritoneal fibroblasts (P-FBs). These cells were isolated from the peritoneal biopsies of children with PD, and these cells had fibroblast-like morphology, and were immunopositive for α-SMA but negative for CK-18, confirming their fibroblast nature (Figure 3). As expected, when P-FBs were treated with PDE, they proliferated progressively and produced high amounts of ECM (Figure 4 and Figure 6). Moreover, in response to the PDGF-B or TGF-β treatment, similarly to FBs of PDE origin, they increasingly proliferated and produced ECM components, respectively.

To investigate the role of PDE-EVs on the behavior of primary P-FBs, first, we demonstrated that our EVs could penetrate into these cells (Figure 3). Moreover, we confirmed that PDE-EVs reduced the PDE or TGF-β treatment-induced ECM production (Figure 6) and the PDE- or PDGF-B-induced proliferation (Figure 4) of the primary P-FBs as well. Thus, our data clearly show that PDE-EVs can inhibit the activation of primary P-FBs in a TGFβ- and/or PDGF-B-dependent manner.

Taken together, our in vitro experiments suggest that PDE—due to its profibrotic factor content—induces the mesenchymal differentiation of primary MCs and the proliferation and ECM production of primary fibroblasts of different origin.

More importantly, our experiments widely demonstrated for the first time that PDE-EVs significantly inhibit the profibrotic effects of PDE in a TGF-β- and/or in a PDGF-B-dependent manner, representing an endogenous mechanism influencing the behavior of the cells involved in the health and pathology of the peritoneum.

In accordance, Pearson and co-workers previously investigated the protein composition of EVs of PDE origin similar then ours [14]. Unfortunately, they did not investigate the biological role of the isolated EVs; however, they identified 3076 cargo proteins. From these cargo proteins, 7 are enriched in the GO term of the “negative regulation of fibroblast proliferation” (GO:0048147—consisting of 29 proteins) and 14 are enriched in the “ECM disassembly” (GO:0022617—consisting of 34 proteins) (Figure 9). The large enrichment of the proteins in the above fibrosis-related GO terms support our observation about the role of PDE-EVs in the regulation of fibrosis (Figure 9).

Finally, we investigated whether the observed in vitro effects of PDE-EVs may also influence peritoneal fibrosis in vivo. In our experiment, the intraperitoneal administration of CG was used to induce peritoneal fibrosis [21]. Similarly to the changes observed in the human peritoneal biopsies of children with PD, the CG treatment induced the mesenchymal dedifferentiation of the mesothelial layer, as demonstrated by the increased presence of mesenchymal and the decreased presence of epithelial markers and the definite submesothelial thickening (Figure 8) [44,45]. As shown by fluorescent microscopy, PDE-EVs penetrated deep into the peritoneum (Figure 7), suggesting their role in the pathology of the peritoneum. Although PDE-EVs were administered only twice during the one-week-long experiment, the thickness of the peritoneum returned to the control level by the end of the experiment on the seventh day of GC treatment (Figure 8). At the same time, the PDE-EV treatment increased the synthesis of the CK-18 epithelial marker of the peritoneum and decreased that of α-SMA and vimentin, indicating the in vivo effect of PDE-EVs on the mesenchymal dedifferentiation of the mesothelial cells (Figure 8). All these observations suggest that the process of EMT, and consequently, the formation of fibroblasts, was inhibited by PDE-EVs in vivo. In accordance with the observed histological or molecular changes, PDE-EVs also improved the peritoneal transport functions, as shown by the increased dextran permeability of the peritoneum on the third day of CG treatment (Figure 8). Although this beneficial effect was transient, it must be noted that the effect of CG is considerable and durable, and the ultrafiltration capacity of the peritoneum does not return to the normal level even two weeks after the last CG treatment (Figure 7). Therefore, it is not expected that the two doses of EV can restore long-term peritoneal functions. In this context, the CG-induced mouse model of peritoneal thickening has some similarities to EPS [46].

Taken together, our data suggest that EVs may have therapeutic potential in preventing peritoneal scarring and EPS formation in vivo. Moreover, due to the similarity of the fibrotic processes of different organs, EVs isolated from the PDE may also be useful in the treatment of other FDs. This observation may have a huge impact on the treatment of FDs, which are responsible for 45% of all death in the developed world [1].

In summary, we demonstrated that EVs derived from the PDEs of children with PD have a strong anti-fibrotic effect, which may represent a new endogenous repair mechanism. We evidenced that PDE-EVs attenuate the EMT of the mesothelial cells and also decrease the activation of fibroblasts of peritoneal origin in vitro and in vivo. These effects of PDE-EVs were independent from the patients from whom they were isolated and related to TGF-β- and PDGF-B-driven processes. Therefore, our data indicate that PDE-EVs may have a previously unrecognized therapeutic potential in the treatment of PD-related peritoneal thickening, and even in that of other FDs.

## Figures and Tables

**Figure 1 cells-13-00605-f001:**
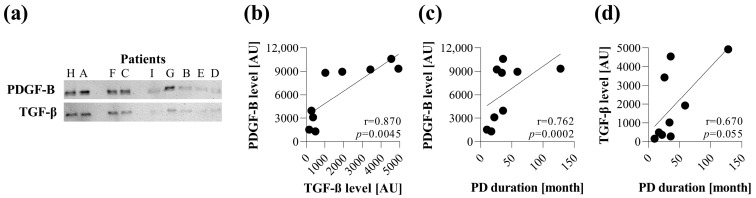
Correlation between the amount of PDGF-B and TGF-ß in the peritoneal dialysis effluent (PDE) and the duration of peritoneal dialysis (PD) in children. The amount of PDGF-B and TGF-ß in the lyophilized PDE samples (n = 9, patients A–I) was measured by Western blot analysis (**a**). Correlation between PDGF-B and TGF-ß levels (**b**), PDGF-B level, and PD duration (**c**), and TGF-ß level and PD duration (**d**) was determined by Spearman coefficients (r). Dots represent individual values. AU: arbitrary units.

**Figure 2 cells-13-00605-f002:**
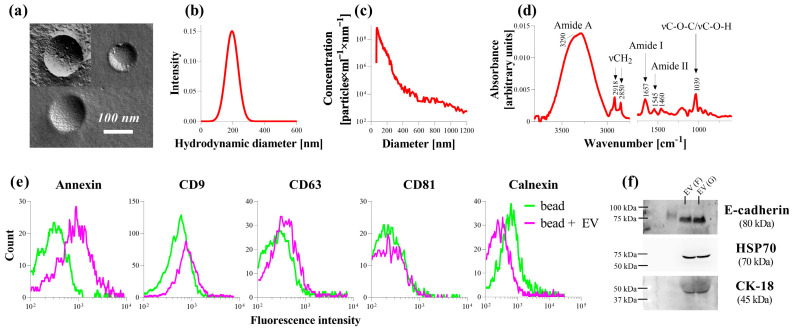
Characterization of peritoneal dialysis effluent (PDE)-derived EVs. Morphology of PDE-EVs of patients A, F, and G was investigated by freeze-fracture combined transmission electron microscopy. Scale bar was 100 um in each case. (**a**). Size distribution and particle number of PDE-EVs of patient G were determined by dynamic light scattering (**b**) and microfluidic resistive pulse sensing (**c**). Protein and lipid content and their ratio from PDE-EVs of patient G was analyzed by Fourier transform infrared spectroscopy (**d**). Presence of AnnexinV, CD9, CD63, CD81, and calnexin was investigated by the flow cytometry on the PDE-EVs of patient G (**e**) and that of E-cadherin, HSP70 and CK-18 was detected by Western blot in the PDE-EVs of patients F and G (**f**).

**Figure 3 cells-13-00605-f003:**
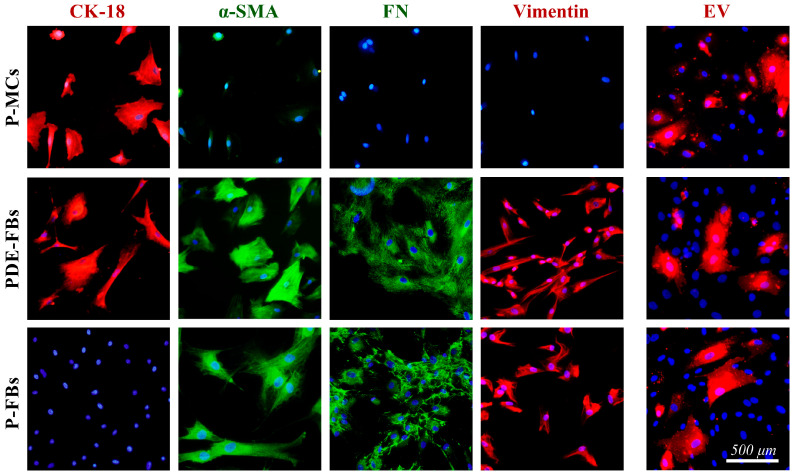
Characterization and EV-uptake of primary cells. The presence of cytokeratin (CK)-18, alpha-smooth muscle actin (α-SMA), fibronectin (FN), and vimentin was examined by immunofluorescent staining on primary peritoneal mesothelial cells (P-MCs), primary fibroblasts isolated from peritoneal dialysis effluents (PDE-FBs), and primary fibroblasts isolated from peritoneal biopsy (P-FBs) of pediatric patients. Internalization of DiI-labeled, peritoneal dialysis effluent-derived EVs of patient G into primary cells was investigated after 24 h of incubation. Cell nuclei were counterstained with Hoechst 33342 (blue).

**Figure 4 cells-13-00605-f004:**
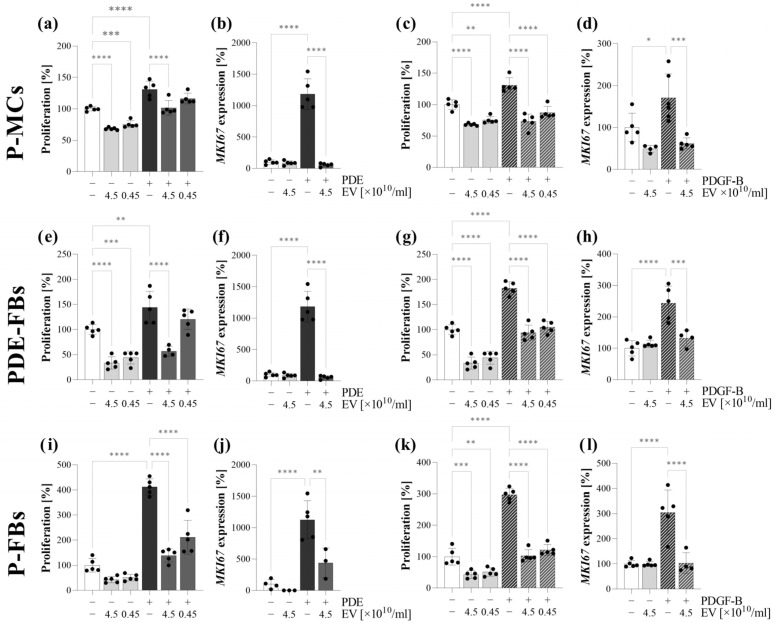
Effect of peritoneal dialysis effluent (PDE)-derived EVs on the proliferation of peritoneal cells. Primary peritoneal mesothelial cells (P-MCs; (**a**–**d**)) of patient J, fibroblast from the PDE of patient G (PDE-FBs; (**e**–**h**)), and fibroblast from the peritoneal biopsy of patient B (P-FBs; (**i**–**l**)) were stimulated by treatment with PDE of patient G (**a**,**b**,**e**,**f**,**i**,**j**) or PDGF-B (**c**,**d**,**g**,**h**,**k**,**l**) in the absence or presence of EVs of patient G. Cell proliferation was measured by MTT assay. The mRNA expression of MKI67 was determined by RT-PCR in comparison with GAPDH as an internal control. The results, normalized to the control group, are presented as the mean ± SD, with dots representing individual values (n = 5). * *p* < 0.05; ** *p* < 0.01; *** *p* < 0.001; **** *p* < 0.0001 (one-way ANOVA with Bonferroni test).

**Figure 5 cells-13-00605-f005:**
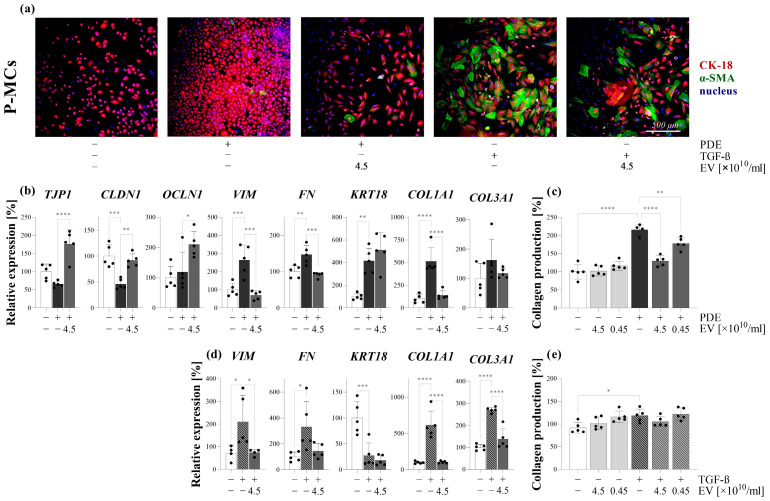
Effect of peritoneal dialysis effluent (PDE)-derived EVs on the mesothelial–mesenchymal transition and collagen production of peritoneal mesothelial cells. Primary peritoneal mesothelial cells (P-MCs) of patient J were stimulated by the treatment with the PDE of patient G (**a**–**c**) or TGF-ß (**a**,**d**,**e**) in the absence or presence of the EVs of patient G. The presence of CK-18 and α-SMA was investigated by immunofluorescent staining (**a**). Cell nuclei were counterstained with Hoechst 33342. The mRNA expression of TJP1, CLDN1, OCLN1, VIM, FN, KRT18, COL1A1, and COL3A1 was determined by RT-PCR in comparison with GAPDH as an internal control (**b**,**d**). Collagen production was measured by the Sirius Red assay (**c**,**e**). The results, normalized to the control group, are presented as the mean ± SD, and dots represent individual values (n = 5). * *p* < 0.05; ** *p* < 0.01; *** *p* < 0.001; **** *p* < 0.0001 (one-way ANOVA with Bonferroni test).

**Figure 6 cells-13-00605-f006:**
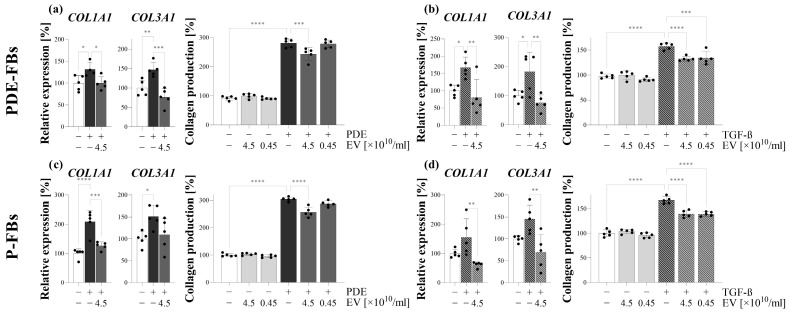
Effect of peritoneal dialysis effluent (PDE)-derived EVs on the collagen production of peritoneal fibroblasts. A primarily peritoneal fibroblast from the PDE of patient G (PDE-FBs; (**a**,**b**)) and a fibroblast from the omental peritoneal biopsy (P-FBs; (**c**,**d**)) of patient B were stimulated by treatment with the PDE of patient G (**a**,**c**) or TGF-ß (**b**,**d**) in the absence or presence of the EVs of patient G and of patient I, respectively. The mRNA expression of COL1A1 and COL3A1 was determined using RT-PCR in comparison to GAPDH as an internal control. Collagen production was measured by the Sirius Red assay. The results, normalized to the control group, are presented as the mean ± SD, and dots represent the individual values (n = 5). * *p* < 0.05; ** *p* < 0.01; *** *p* < 0.001; **** *p* < 0.0001 (one-way ANOVA with Bonferroni test).

**Figure 7 cells-13-00605-f007:**
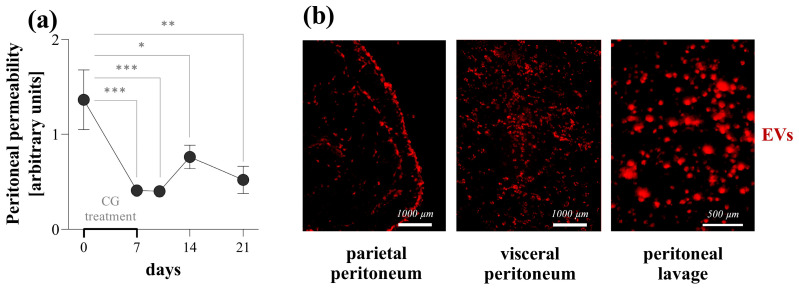
Experimental model to investigate the effect of EVs in peritoneal fibrosis. In the in vivo experiments, mice were treated with daily intraperitoneal (i.p.) injections of chlorhexidine digluconate (CG) for 7 days to induce peritoneal fibrosis. The peritoneal permeability was measured by determining the ratio of 70 kDa/4 kDa fluorescent dextrans in the serum, administered i.p. 30 min before blood sampling (**a**). Results are presented as the mean ± SD (n = 6). * *p* < 0.05; ** *p* < 0.01; *** *p* < 0.001 (one-way ANOVA with Bonferroni test). The penetration of DiI-labeled peritoneal dialysis effluent-derived EVs of patient F in the peritoneal tissue was studied in healthy mice by the immunofluorescent microscopy of parietal peritoneum, visceral peritoneum, and peritoneal lavage samples, 3 days after the i.p. administration of EVs (**b**).

**Figure 8 cells-13-00605-f008:**
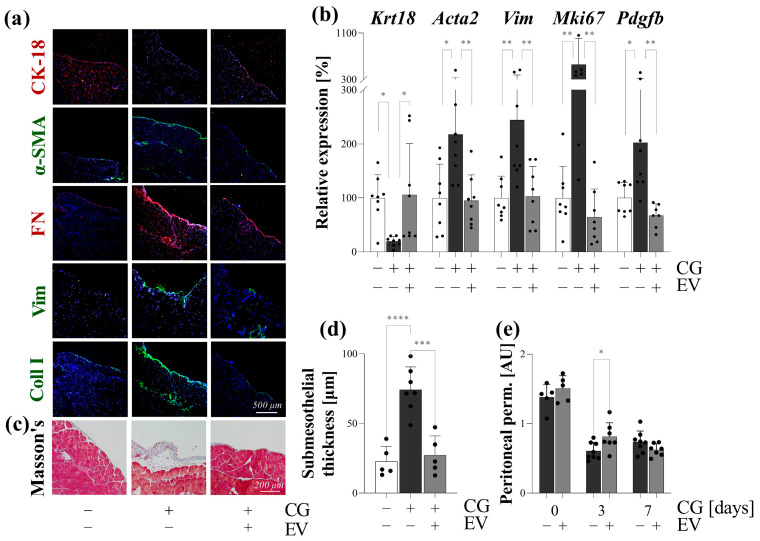
Effect of peritoneal dialysis effluent (PDE)-derived EVs on peritoneal fibrosis. In the in vivo experiments, mice were treated with daily intraperitoneal (ip) injections of chlorhexidine digluconate (CG) for 7 days with or without the EV of patient F co-treatment on days 1 and 4. Expression levels of CK-18, α-SMA, FN, vimentin (Vim), and collagen type I (Coll I) were evaluated by immunofluorescent staining on parietal peritoneal sections (**a**). Cell nuclei were counterstained with Hoechst 33342 (blue). The mRNA expressions of Krt18, Acta2, Vim, Mki67, and Pdgfb was determined by RT-PCR in comparison with Gapdh and normalized to the control group (**b**). Submesothelial thickness was investigated on Masson’s trichrome-stained peritoneal sections of mice (**c**) and quantified by graphical analysis (**d**). Peritoneal permeability was measured by determining the ratio of 70 kDa/4 kDa fluorescent dextrans in the serum, administered i.p. 30 min before blood sampling (**e**). Results are presented as mean ± SD (n = 5–8). * *p* < 0.05; ** *p* < 0.01; *** *p* < 0.001; **** *p* < 0.0001 (one-way (**b**,**d**) or two-way (**e**) ANOVA with Bonferroni test). AU: arbitrary unit.

**Figure 9 cells-13-00605-f009:**
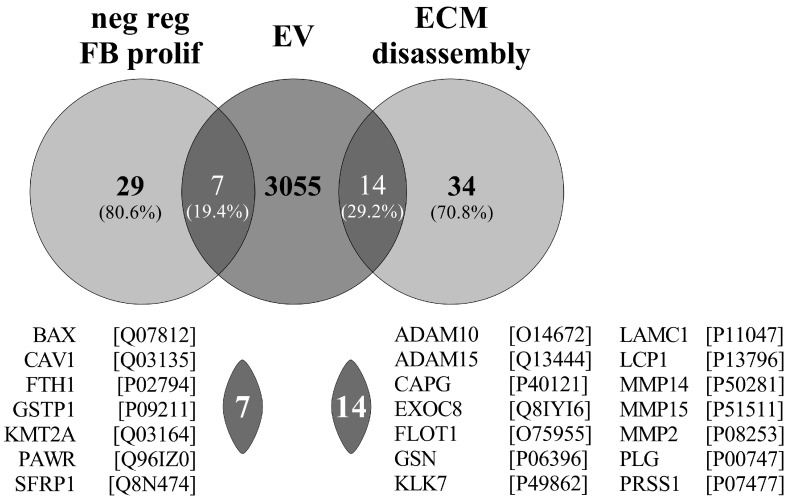
Intersectional analysis of EV-related proteins and antiproliferative/antifibrotic molecules. Venn diagrams show the number of proteins detected in peritoneal dialysis effluent (PDE)-derived EVs and involved in “negative regulation of fibroblast proliferation” (GO:0048147) and “ECM disassembly” (GO:0022617) biological processes based on Gene Ontology database. ECM: extracellular matrix, FB: fibroblast, EV: extracellular vesicle.

**Table 1 cells-13-00605-t001:** Main clinical features and samples of the enrolled patients. Abbreviations: PDE—peritoneal dialysis effluent; EV—extracellular vesicles isolated from PDE; P-MC—human primary mesothelial cells; PDE-FB—human primary fibroblast from PDE; P-FB—human primary fibroblast from peritoneal biopsies; F—female; M—male.

Patient	Age (Years)	Sex	Disease Group	PD Duration (Months)	Samples	Primary Cells
A	9.1	F	Kidney and urinary tract malformations/congenital kidney diseases	36	PDE, EV	PDE-FB
B	1.9	M	Cortical/tubular necrosis	22	Peritoneal biopsy	P-FB
C	5.7	M	Cortical/tubular necrosis	59	PDE	
D	18.1	M	Kidney and urinary tract malformations/congenital kidney diseases	17	PDE	
E	17.7	M	Glomerulopathy	10	Peritoneal biopsy	P-FB
F	13.9	M	Metabolic diseases	26	PDE, EV	PDE-FB
G	3.2	M	Cortical/tubular necrosis	34	PDE, EV	PDE-FB
H	22.1	F	Kidney and urinary tract malformations/congenital kidney diseases	128	PDE	
I	32.3	M	Kidney and urinary tract malformations/congenital kidney diseases	36	PDE, EV	PDE-FB
J	14	M	Kidney and urinary tract malformations/congenital kidney diseases	0	Peritoneal biopsy	P-MC

**Table 2 cells-13-00605-t002:** Nucleotide sequences and species specificity of the primer pairs applied for the real-time reverse transcriptase polymerase chain reaction (RT-PCR).

Gene	Primer Pairs		Species
*CLDN1*	F:	5′-TAT GCC GGC GAC AAC ATC GTG ACC-3′	Human
	R:	5′-TCC CAG GAG GAT GCC AAC CAC CAT-3′	
*COL1A1*	F:	5′-CTG CCC GGG CGC CGA AGT C-3′	Human
	R:	5′-CCC TCG ACG CCG GTG TTT CTTG-3′	
*COL3A1*	F:	5′-GTC CCC TGG CTC AAA TGG CTC AC-3′	Human
	R:	5′-GGG GCC CCT TGC TCC TAT TAG TCC-3′	
*FN*	F:	5′-GGC TGC CCA CGA GGA AAT CTG C-3′	Human
	R:	5′-GTG CCC CTC TTC ATG ACG CTT GTG-3′	
*GAPDH*	F:	5′-AGC AAT GCC TCC TGC ACC ACC AA-3′	Human
	R:	5′-GCG GCC ATC ACG CCA CAG TTT-3′	
*KRT18*	F:	5′-AGC GCC AGG CCC AGG AGT ATG AGG-3′	Human
	R:	5′-TAT CCG GCG GGT GGT GGT CTT TTG-3′	
*OCLN1*	F:	5′-TGA ATG ACA AGC GGT TTT ATC CAG-3′	Human
	R:	5′-TGA AGT CAT CCA CAG GCG AAG TTA-3′	
*MKI67*	F:	5′-CCC CTA CGG ATT ATA CTC AAC TTA-3′	Human
	R:	5′-TGT AAT ATT GCC TCC TGC TCA-T-3′	
*TJP1*	F:	5′-ACC ACA AGC GCA GCC ACA ACC AAT-3′	Human
	R:	5′-GGG GTG GGC TCC TCC AGT CTG ACA T-3′	
*VIM*	F:	5′-GAG GCT GCC AAC CGG AAC AAT GAC-3′	Human
	R:	5′-TCC TGC AGG CGG CCA ATA GTG TCT-3′	
*Acta2*	F:	5′-CCC CTG AAG AGC ATC GGA CA-3′	Mouse
	R:	5′-TGG CGG GGA CAT TGA AGG T-3′	
*Gapdh*	F:	5′-ATC TGA CGT GCC GCCTGGAGAAAC-3′	Mouse
	R:	5′-CCCGGCATCGAAGGTGGAAGAGT-3′	
*Krt18*	F:	5′-GCC TTG CCG CCG ATG ACT TTA GA-3′	Mouse
	R:	5′-TCC AGC TGC AGC CTT GTG ATG TTG-3′	
*Mki67*	F:	5′-GGC GGC GAC GAC CCA TTC-3′	Mouse
	R:	5′-TGG ATG TGG TAG CCG TTT CTC AGG-3′	
*Pdgfb*	F:	5′-CTG GGC GCT CTT CCT TCC TCT C-3′	Mouse
	R:	5′-CCA GCT CAG CCC CAT CTT CAT C-3′	
*Vim*	F:	5′-CTG CTG CCC TGC GTG ATG TG 3′	Mouse
	R:	5′-TGG CGC TCC AGG GAC TCG T-3′	

## Data Availability

Data are contained within this article.

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
