# Peer review of "Extracellular Vesicles of Patients on Peritoneal Dialysis Inhibit the TGF-β- and PDGF-B-Mediated Fibrotic Processes"

_cells, 2024, doi:10.3390/cells13070605_

Round 1
Reviewer 1 Report
Comments and Suggestions for Authors
This paper offers a good direction to explore the role of extracellular vesicles (EVs) in peritoneal fibrosis. Some questions and suggestions are raised for your consideration.
1. The introduction currently lacks a clear statement of the specific aims of the study, particularly in relation to the identified gaps in the literature. A final paragraph in the introduction that outlines the objectives, methodologies, and how these approaches address the existing gaps would be helpful.
2. The resolution and clarity of the images in Figure 2a need improvement. Are all images presented with the same scale bar?
Regarding sections 3.4 – 3.6:
1. What was the rationale behind choosing a concentration of 4.5 X 10^10 particles for the EVs?
2. Related to the above, how much PDE was processed to obtain this specific amount of EVs? Was there a noticeable variance in the concentration of EVs derived from the PDE of different patients?
3. While PDE appears to promote proliferation, mesenchymal transition, and collagen production, PDE-EVs seem to inhibit these processes. The author should elaborate on these divergent effects.
Author Response
Dear Reviewer,
Thank you very much for taking time to review this manuscript. Please find the detailed responses below and the attached new version of the manuscript.

Reviewer 2 Report
Comments and Suggestions for Authors
In this paper, Szebeni et al. described the effects of PDE-EVs to peritoneal cells and found that EVs inhibited the proliferation and production of extracellular matrix. Their work showed the potential of PDE derived EVs for therapy of peritoneal fibrosis. They also described the effects of each patient’s EV for proliferation in supplementary figures. The manuscript was well described. I have some points for discussion.
1. The authors described which patient’s EV they used in supplementary figure 1. I suggest this information should be written in figure legends in each figure.
2. They did many experiments for proliferation with different patient’s EVs. I think at least for inhibition of proliferation, authors should describe how general the effect of PDE-EVs on proliferation in discussion.
3. The authors described the inhibitory effects of patients’ EVs for peritoneal fibrosis. However, patients still became peritoneal fibrosis despite they have their therapeutic EVs. The authors should discuss about how to improve the therapeutic ability of EVs.
Author Response
Dear Reviewer,
Thank you very much for taking the time to review this manuscript. Please find the detailed responses below, the corresponding revisions are highlighted.

Reviewer 3 Report
Comments and Suggestions for Authors
A common hallmark of fibroproliferative diseases is the activation of fibroblasts and the excessive accumulation of the extracellular matrix, leading to the destruction of the healthy architecture of the affected organ and finally to the decline of its function.
One this example is peritoneal fibrosis which is developed in more than 50%-80% of patients subjected to peritoneal dialysis. The peritoneal fibrosis leads to progressive submesothelial thickening and impaired peritoneal filtration making it necessary to change the modality of the renal replacement therapy. In this study, Szebeni et al., have investigated the potential of extracellular vehicles (EVs) of patients on peritoneal dialysis as a potential therapeutic possibility for delaying or even reversing peritoneal fibrosis.
They have tried to address the impact of peritoneal dialysis effluent-EVs on the epithelial-mesenchymal transition and collagen production of the peritoneal mesothelial cells and fibroblast both in vitro and in vivo, using chlorhexidine digluconate induced mice model of peritoneal fibrosis. Despite the intriguing approaches and the potential interest in the authors´ hypothesis some additional studies and controls are needed to substantiate some of the claims of the authors, and certain aspects should be clarified especially those directly related to EVs origin, characterization, and content.
Using various methods, the authors have characterized the PDE-derived EVs (from Patients A, F, G, and I) in Figure 2. Using flow cytometry, the presence of annexin V, CD9, CD63, CD81, and calnexin has been examined in panel D. It's unclear from the profile if it comes from the purified sample taken from Patient A, F, G, or I, or if it's an average of all the study samples. The figure legend ought to reflect this. Furthermore, the western blot of the four purified EVs is shown in panel f, which is denoted as g in the figure legend (this should be corrected). Included should be the label of the specific sample that was placed into each well.
Why do the samples appear to differ in terms of E-cad, hsp70, and CK18 expression? Including other markers like Tsg101 and Alix or performing a western blot on some of the ones examined by flow cytometry would be nice to thoroughly verify that the purified EV levels are consistent across the board.
An additional question pertains to whether the quantity and composition of EVs vary based on the duration of peritoneal dialysis treatment for the patients. This should be discussed.
The following performed experiments nicely show that EVs produced from PDE have a potent antifibrotic activity in vitro, which is linked to reduced activation of fibroblasts from diverse origins and inhibited EMT of the mesothelial cells. The exact origin of these EVs is yet unknown, though. Could the authors add any information on this point? Is it feasible to evaluate the expression of some specific markers to clarify the possible origin?
The process by which these EVs prevent the effects of TGF and PDGF should also be better explained. Could these EVs be functioning as a mechanism for sequestering these factors? If yes, how is this even possible?
Despite not conducting a proteomic analysis on their purified EVs, the authors have assessed the GO enrichment analysis of the PDE-EV proteins in Figure 9. This analysis was based on the prior proteomic data of Pearson and colleagues, and they identified 7 proteins as belonging to the GO term "negative regulation of fibroblast proliferation" and 14 proteins in the GO term "ECM disassembly." Considering this, the authors should analyse by western blot the expression of a few of the hits found in the proteome analysis in their purified PDE-EVs.
Author Response

(The authors gave the same response as above.)

Reviewer 4 Report
Comments and Suggestions for Authors
In this contribution, the authors focus on extracellular vesicles in peritoneal dialysis. It is a relevant topic and fits the scope of the MDPI Cell journal.
I do not have major comments on the main part of the manuscript (in vivo/vitro tests), and I find it well-written and clear. However, I am not very satisfied with the part related to the characterisation of EVs.
(1) My main points are related to the Material and Methods part related to the characterisation of EVs. It is written in a generic (i.e. in an unreproducible) way.
For example:
Line 161: The sample was diluted 10-fold… - Which sample? After SEC? What was the particle (EV) concentration? Or at least protein content?
Line 169: Any information about the measured sample is given for DLS. Same as above.
Line 173: FTIR: Again, the same issue is present. “5µL of each sample…” Which sample? After SEC? What was the particle (EV) concentration? Or at least protein content?
Line 203: 100 µL EVs… Again, we do not know anything about the measured/labelled sample. What was the ratio between EVs and dye?
Please add this information to all analytical methods used.
(2) The statement (Line 138-140) “The PBS soluble fractions were collected and analyzed by the following methods according to the International Society for Extracellular Vesicles (ISEV) recommendations to demonstrate and validate the EV content of the samples.”
If you claim this, you should fulfil the MISEV criteria. It is not just a MUST sentence in the EV article. First, you must provide experimental details. Nota bene: you even omitted the reference to these guidelines. I strongly recommend rewriting the M&M section according to the recently published guidelines.
Welsh, J. A., Goberdhan, D. C. I., O’Driscoll, L., Buzas, E. I., Blenkiron, C., Bussolati, B., Cai, H., Di Vizio, D., Driedonks, T. A. P., Erdbrügger, U., Falcon-Perez, J. M., Fu, Q.-L., Hill, A. F., Lenassi, M., Lim, S. K., Mahoney, M. G., Mohanty, S., Möller, A., Nieuwland, R., … Witwer, K. W. (2024). Minimal information for studies of extracellular vesicles (MISEV2023): from basic to advanced approaches. Journal of Extracellular Vesicles, 13, e12404. https://doi.org/10.1002/jev2.12404
(3) If the EVs are Annexin A5, CD9, CD63 positive and CD81, Calnexin negative. Can you be sure that these EVs are not platelet-derived? What about CD41?
(4) Please check the proper format of characters in the FTIR part (lines 381-398). I find the sentence (Line 388-391) “copy-paste” from your previous manuscript. Please reformulate this sentence. In this contribution, you did not introduce the P/L ratio (you already established it a few years ago). It does fit well here. It can be shortened, which will make the paragraph more straightforward. Also, I will remove the labelling of the nonessential peak in the FTIR spectra (e.g. 1738 cm-1) for better readability.
Comments on the Quality of English LanguageMinor editing and formatting are required.
Author Response

(The authors gave the same response as above.)

Round 2
Reviewer 1 Report
Comments and Suggestions for Authors
No further comments.
Reviewer 3 Report
Comments and Suggestions for Authors
I really appreciate the authors' efforts to address my different comments and concerns. The revision clarifies parts of the points, but some cannot be assessed because the original patients´ EVs cannot be purified again as these patients have already undergone transplantation.
In any case, based on the interest of the manuscript and the provided data, I consider that the manuscript should be accepted for publication.